# Advances in Immunotherapy for Hepatocellular Carcinoma

**DOI:** 10.3390/cancers15072070

**Published:** 2023-03-30

**Authors:** Satoru Hagiwara, Naoshi Nishida, Masatoshi Kudo

**Affiliations:** Department of Gastroenterology and Hepatology, Kindai University Faculty of Medicine, Osaka 589-8511, Japan

**Keywords:** hepatocellular carcinoma, immune checkpoint inhibitors, tumor immune environment

## Abstract

**Simple Summary:**

Immunotherapy for hepatocellular carcinoma (HCC) is progressing rapidly. In particular, analyses of the tumor immune environment in HCC are progressing. Moreover, custom-made immunotherapy is being considered. A new taxonomy of immune subclasses of HCC is described. A relationship between etiology and immunotherapy has also been suggested. Furthermore, interesting reports have been made on the relationship between nonalcoholic steatohepatitis-HCC and immunotherapy. Understanding the tumor immune environment is essential for implementing immune checkpoint inhibitors (ICIs) for hepatocellular carcinoma. It is also important in managing the side effects of ICIs as well. The purpose is to summarize the progress of immunotherapy in hepatocellular carcinoma and to utilize it for future treatment.

**Abstract:**

Immune checkpoint inhibitors (ICIs) aim to induce immune responses against tumors and are less likely to develop drug resistance than molecularly targeted drugs. In addition, they are characterized by a long-lasting antitumor effect. However, since its effectiveness depends on the tumor’s immune environment, it is essential to understand the immune environment of hepatocellular carcinoma to select ICI therapeutic indications and develop biomarkers. A network of diverse cellular and humoral factors establishes cancer immunity. By analyzing individual cases and classifying them from the viewpoint of tumor immunity, attempts have been made to select the optimal therapeutic drug for immunotherapy, including ICIs. ICI treatment is discussed from the viewpoints of immune subclass of HCC, Wnt/β-catenin mutation, immunotherapy in NASH-related HCC, the mechanism of HPD onset, and HBV reactivation.

## 1. Introduction

In recent years, various molecular-targeted drugs have become available for treating hepatocellular carcinoma, contributing to improved prognosis in advanced hepatocellular carcinoma [1,2,3]. However, many therapeutic agents using molecular-targeted drugs often develop tumors due to the acquisition of resistance mutations during their course [4,5,6]. Drugs that easily target intracellular molecules acquire resistance due to structural changes in target molecules due to the accumulation of mutations during genome replication in cancer cells.

On the other hand, immunotherapy, including immune checkpoint inhibitors (ICIs), aims to modify the immune response against tumors. Furthermore, the target molecules or targets are immune system cells or molecules expressed therein. Since these are mainly expressed in noncancer cells, they are less likely to develop resistance compared to molecular-targeted drugs and are characterized by long-lasting antitumor effects in response to treatment [7]. In addition, since ICIs have a mechanism of action different from that of molecular-targeted therapeutics, they are expected to be effective in cases where molecular-targeted therapeutics are ineffective or in which resistance is acquired. Immune checkpoint inhibitors are expected to be more effective in tumors exhibiting genomic instability that produce various tumor antigens [8,9]. Furthermore, it is also attracting attention as a drug that complements the weaknesses of molecular-targeted therapeutics. However, since the therapeutic effect depends on the tumor’s microimmune environment [10], it is essential to understand the immune environment of hepatocellular carcinoma for the selection of therapeutic indications for ICIs and the development of biomarkers.

### 1.1. Immune Subclass of Hepatocellular Carcinoma (Reported to Date)

#### 1.1.1. Inflamed Class and Noninflamed Class

A classification of the “immune microenvironment” in HCC was reported by Sia et al. in 2017 [11,12]. Based on the results of gene expression analysis of 956 HCC cases, they defined the “immune class” (hereafter referred to as the “inflamed class”). Furthermore, they reported that about 25% of HCCs were of the inflamed class, characterized by increased programmed cell death protein 1 (PD-1) and programmed death ligand-1 (PD-L1) expression. The inflamed class is further divided into two subclasses, the immune active subclass and the immune exhausted subclass. The first immune active subclass is rich in interferon (IFN) signals. The second subclass, the immune exhausted subclass, has increased transforming growth factor beta (TGF-β) signaling activity and T cell exhaustion-related gene expression.

On the other hand, about 60–80% of HCCs are noninflamed classes. They are divided into two groups based on the immune escape mechanism. The first is the intermediate class, and the other is the excluded subclass characterized by Catenin Beta 1 (CTNNB1) mutation, which is poor in immune cell infiltration.

#### 1.1.2. *CTNNB1* Mutation and Immune Exclusion

According to the molecular classification of HCC, *CTNNB1* mutations are classified as a nonproliferative subclass with a favorable prognosis. However, on the other hand, as mentioned above, they are considered a subclass that is unlikely to cause immune cell infiltration. Catenin Beta 1 mutations are known to be involved in Wnt/β-catenin pathway activation. In 2019, an immune evasion mechanism in Wnt/β-catenin pathway activation was reported using HCC model mice [13]. Harding et al. also clinically demonstrated the negative impact of Wnt/β-catenin pathway activation on disease control rate/progression-free survival in HCC patients using ICIs [14,15]. The phosphorylation of multiple serine/threonine residues in exon 3 of CTNNB1 by GSK3α/β results in its recognition by the E3 ubiquitin ligase β-TrCP and rapid proteasomal degradation. In CTNNB1-mutant hepatocellular carcinoma, β-catenin escapes degradation by the ubiquitin-proteasome system and accumulates in the nucleus to form a complex with TCF/LEF. This causes the target gene to be expressed. It is also known that CTNNB1 and TP53 mutations are mutually exclusive.

### 1.2. Proposing New Immune Subclasses 

#### 1.2.1. Inflamed Class

In recent years, new immune subclasses have been proposed (Figure 1). Montironi and Llovet et al. analyzed tumors and matched nontumor areas in 240 HCC patients [16,17]. They performed immune cell infiltration, intratumoral lymphocyte infiltration, stromal tumor-infiltrating lymphocytes, immunohistochemical staining (LAG-3, TIM-3, CTLA-4, and TUGIT), multiple immunofluorescence analyses (PD-1, PD-L1, and CD8) and gene analyses (RNA sequencing, TCR sequencing, and whole-exon sequencing), and an immune subclass analysis. They reported that they could capture up to 90% of the inflamed class using a 20-gene signature. Applying this gene signature to other cohorts using ICIs for advanced HCC showed significant upregulation in responders and was reported to be useful for ICI treatment effects.

#### 1.2.2. New Subclass of Inflamed Class (Proposal of Immune-like Subclass)

The inflammatory class is said to be divided into three subclasses: (1) the immunoreactive subclass, (2) the immunodepletion subclass, and (3) the immune-like subclass. In other words, the immune active and immune exhausted subclass have been reported as subclasses of the inflamed class. In addition to this, we have proposed a new immune-like subclass. High IFN signaling, cytokines, and a diverse T cell repertoire characterize the immune-like subclass. Specifically, immune checkpoint molecules, including Cytotoxic T-lymphocyte antigen 4, PD-1, and PD-L1, were significantly overexpressed in the immune-like subclass. In addition, we also observed increased IFN signals, cytolytic activity, and inflammation-related pathways related to the response to immunotherapy.

#### 1.2.3. Noninflamed Class and Its Features

The noninflamed class is divided into (1) an intermediate subclass and (2) excluded subclass based on the mechanism of immune evasion. The intermediate subclass has an intermediate degree of immune cell infiltration between the inflamed class and the excluded subclass. It is often associated with tumor protein 53 (*TP53*) mutations. Furthermore, it is characterized by high levels of chromosomal instability. Additionally, it is associated with the deletion of genes associated with antigen presentation and IFN signaling. The excluded subclass has the lowest immune cell infiltration into tumors and a higher frequency of *CTNNB1* mutations than the intermediate subclass [16,17].

### 1.3. Duality of Wnt/β-Catenin Mutations

Previously, research results on Wnt/β-catenin mutations in HCC had different aspects. In other words, while the phenotype with less invasion/metastasis and good prognosis accounts for the majority, there are also phenotypes with more invasion/metastasis and poor prognosis [18]. A clue to understanding this duality is the presence of a target gene for Wnt/β-catenin signaling (Hepatocyte nuclear factor 4 alpha *(HNF4α*)), as well as Forkhead box protein M1 (*FOXM1*), which influences Wnt/β-catenin signaling and is involved in tumor malignancy. Forkhead box protein M1 induces genes involved in the cell cycle and dedifferentiation in conjunction with β-catenin. Wnts originally regulate undifferentiated cancer stem cells [19]. Colorectal, lung, and breast cancer with Wnt/β-catenin gene mutations cause invasion and metastasis, resulting in poor prognosis. On the other hand, HCC has a good prognosis phenotype with little invasion and metastasis. One factor that explains this discrepancy is that the HCC-specific tumor suppressor gene *HNF4α* is a target gene for Wnt/β-catenin signaling (Figure 2. Modified from [18])

#### 1.3.1. Good Prognosis Group for Wnt/β-Catenin

Tumor suppressor gene *HNF4α*, a member of the steroid hormone receptor superfamily, is specifically expressed in the liver, kidney, and intestine but not in other organs. Hepatocyte nuclear factor 4 alpha is a master gene that regulates the transcription of hepatocyte differentiation, proliferation, morphogenesis, and cell adhesion, amongst others. Furthermore, it negatively regulates *Snail1,2*, which is associated with epithelial-mesenchymal transition (EMT). It has also been reported to promote the cell membrane localization of β-catenin and suppress oncogenes such as cyclin-D1 [18]. It induces differentiation into HCC with a low alpha-fetoprotein (AFP) level, which is less susceptible to invasion and metastasis. This phenotype overlaps well with HCC with a favorable prognosis characterized by cholestasis, previously called green hepatoma [20].

#### 1.3.2. Poor Prognosis of Wnt/β-Catenin

On the other hand, stimulation from some Wnts induces the nuclear translocation of *FOXM1* in other carcinomas. Additionally, *FOXM1* influences Wnt/β-catenin signaling in conjunction with β-catenin and is involved in tumor malignancy. Forkhead box protein M1 promotes the expression of *c-Myc* and *Cyclin-D1* and contributes to the differentiation into HCC with poor prognosis through dedifferentiation into cancer stem cells, promotion of EMT, and neovascularization [18]. This phenotype is characterized by elevated AFP levels, positive cancer stem cell markers (cytokeratin 19, epithelial cellular adhesion molecule, sal-like protein 4), and expression of EMT-related genes. Against this background, it is clinically characterized by being susceptible to invasion and metastasis. When examining the nuclear accumulation of β-catenin and prognosis in a target population containing poorly differentiated HCC of less than 3 cm, the nuclear accumulation of β-catenin is negatively correlated with mortality outcome in well-differentiated HCC. On the other hand, the nuclear accumulation of β-catenin has been reported to be a significant prognostic factor in poorly differentiated HCC [21].

### 1.4. Gadoxetic Acid-Enhanced Magnetic Resonance Imaging (EOB-MRI) as an Imaging Biomarker for Wnt/β-Catenin Mutations

As mentioned above, Wnt/β-catenin mutations influence the therapeutic efficacy of ICIs. It is impossible to perform a tumor biopsy on every patient or every nodule. Therefore, it is urgent to establish diagnostic imaging that predicts the therapeutic effect of ICIs. It has been reported that in the activated state of Wnt/β-catenin signaling, it can be visualized as a hyperintensity nodule in the hepatocellular phase of EOB-MRI [22,23]. We found that high signals in the hepatocellular phase of EOB-MRI indicated treatment resistance in patients treated with ICI monotherapy for HCC [24,25]. On the other hand, there are cases in which there is no relationship between the high signal in the hepatocyte phase of EOB-MRI and the therapeutic effect of ICIs. The transcription factor HNF4α, which plays a role in maintaining mature hepatocyte function, also induces the expression of organic anion transporting polypeptide 1B3 (OATP1B3) [26]. A decrease in HNF4α during HCC dedifferentiation may lead to decreased expression of OATP1B3 regardless of Wnt/β-catenin mutations, resulting in loss of uptake during the hepatocyte phase of EOB-MRI. There may be a discrepancy between EOB-MRI hepatocyte-phase uptake and Wnt/β-catenin mutations in some nodules considered in this light.

These studies are only studies of ICI monotherapy. Therefore, further studies are necessary in the future when a VEGF inhibitor is desired to be used in combination.

### 1.5. Effects of Immunotherapy on Nonalcoholic Steatohepatitis (NASH)-HCC

There are reports of limited therapeutic efficacy of immunotherapy for hepatocellular carcinoma with NASH background (NASH-HCC) [27,28]. Normally, CD8+ T cells recognize antigens presented by major histocompatibility complex type 1 and become cytotoxic T cells (CTL), which recognize cancer antigens via T cell receptors (TCR) to attack. However, NASH-accumulating CD8+ T cells differ from cancer antigen-specific CD8+ T cells. CXC chemokine receptor 6+CD8+ T cells in NASH react with the acetic acid of NASH hepatocytes and attack NASH hepatocytes, showing no reaction to NASH hepatoma. Therefore, under such a tumor immune environment, ICI treatment has been reported to have a limited effect on HCC while possibly inducing the risk of progression of background hepatic fibrosis. Molecular features in NASH-HCC have also been reported. The frequency of Activin receptor type-2A mutations was higher in NASH-HCC than in non-NASH-HCC. In NASH-HCC, β-catenin mutation accounted for 28% and 42%, including related mutations [29]. Furthermore, among NASH-HCC, noncirrhotic NASH-HCC has been reported to have a higher tumor mutation burden than cirrhotic NASH-HCC [30]. Further investigation is desired in the future on the relationship between such gene mutations in NASH-HCC and the effect of ICI treatment.

On the other hand, it should be noted that the above reports are only reports on ICIs alone and not on atezolizumab/bevacizumab combination therapy, which is currently the mainstream of HCC treatment. In other words, vascular endothelial growth factor (VEGF) inhibition has been reported to have various effects, such as the promotion of differentiation into mature dendritic cells for tumor immunity, the promotion of T cell infiltration into the tumor, changes in the intratumoral microenvironment, and the regulation of myeloid-derived suppressor cells and Tregs [31,32,33,34]. It has been suggested that combining ICIs and VEGF inhibitors may reset the HCC therapeutic effect by etiology.

### 1.6. Does Atezolizumab and Bevacizumab Combination Therapy Transform Immune Cold Tumors into Immune Hot Tumors?

In the cancer immune cycle, VEGF prevents the infiltration of circulating CD8-positive T cells into the tumor [34]. The main factors are that VEGF induces FAS ligand and causes CTL to undergo apoptosis [31,32,33,34]. Furthermore, CTL adhesion to the vascular endothelium is reduced by reducing adhesion factors such as intercellular adhesion molecule 1 (ICAM-1) and vascular cell adhesion molecule 1 (VCAM-1). It is known to reduce the migration ability to tumors [35]. Furthermore, blocking the release of chemokines such as CXC chemokine ligand (CXCL) 9, 10, and 11, which are ligands of CXC chemokine receptor 3 (CXCR3), from intratumoral dendritic cells also prevents CTLs from adhering to the vascular endothelium and inhibits intratumoral invasion [36,37]. In particular, this is a frequent phenomenon in immune cold tumors. In contrast, the administration of anti-VEGF antibodies suppresses CTL apoptosis and improves the decrease in ICAM-1 and VCAM-1, thereby enhancing CTL adhesion to the vascular endothelium and migration [32]. In addition, CXCL9, a ligand of CXCR3 that promotes adhesion to vascular endothelium for CXCR3, promotes the release of IFN-ɤ from CD8-positive T cells in the tumor by anti-VEGF antibody and anti-PD-1 antibody [37]. As a result, CXCL9 and ten from dendritic cells anchor CTLs to the vascular endothelium via CXCR3 in the bloodstream, effectively inducing CTL infiltration into the tumor [38,39,40]. Such a mechanism suggests that atezolizumab and bevacizumab combination therapy may transform immune cold tumors into immune hot tumors [41] (Figure 3. Modified from [41]).

### 1.7. Onset Mechanism of Hyperprogression Disease in ICI Treatment

The clinical application of ICIs has revealed that rapid tumor growth, called importance hyperprogressive disease (HPD), occurs in some patients. Programmed cell death protein 1-positive Tregs are important as a pathogenesis mechanism of HPD. Tregs are CD4-positive T cells that express forkhead box protein P3 as a master transcription factor and have immunosuppressive activity [42]. In the cancer microenvironment, Tregs produce inhibitory cytokines such as interleukin (IL) 10 and TGF-β, as well as antagonistic and inhibitory effects of CTLA4 on CD80/CD86-CD28-mediated TCR costimulation [43]. This observation suppresses the function of CD8-positive cells. Kamada et al. reported that in patients with advanced gastric cancer who developed HPD due to PD-1/PD-L1 inhibitors, many PD-1-positive Tregs infiltrated the local tumor area before treatment [44]. Furthermore, we have also reported that PD-1 expression in tumor local Tregs is a useful biomarker for PD-1/PD-L1 inhibitor therapy. In other words, the therapeutic effect depends on the expression balance of CD8-positive T cells and PD-1 on Tregs [44], and it is possible that HPD develops. The mechanism by which PD-1 is expressed on Tregs has also been investigated. Tumors with high expression of PD-1 on Tregs were shown to have high expression of glycolysis-related genes. In particular, the lactate transporter, monocarboxylate transporter 1 (MCT1) is highly expressed. Furthermore, lactate uptake, the final metabolite of tumor cell glycolysis, enhances PD-1 expression [45]. These results suggest that the modification of lactate accumulation in the tumor environment or the inhibition of MCT1 by Tregs may be options for overcoming PD-1 inhibitor resistance.

### 1.8. Hepatitis B (HBV) Reactivation by ICIs

There are several reports of HBV reactivation by ICIs [46]. Zhang et al. reported HBV reactivation from hepatitis B surface antigen (HBsAg)-positive patients upon administration of anti-PD-1/PD-L1 antibodies [47]. Of the 114 HBsAg-positive patients, 79 were HBV DNA-negative. Reactivation occurred in 1 of 55 (1.8%) who received prophylactic antiviral therapy and 5 of 24 (20.8%) who did not receive prophylactic antiviral therapy. Additionally, of 511 HBsAg-positive patients, 2 of 464 (0.4%) received prophylactic antiviral therapy, and 3 of 47 (6.4%) did not receive prophylactic antiviral therapy. In total, 5 cases (0.9%) had reactivation [48]. The two cases in which reactivation was observed despite prophylactic antiviral therapy included one in which esophageal varices had ruptured, and it was difficult to continue taking antiviral drugs. In the other case, oral compliance was poor. In other words, all cases that caused HBV reactivation were cases in which prophylactic antiviral therapy could not be introduced or continued. On the other hand, they also reported that none of the 2954 HBsAg-negative cases had reactivation. These reports indicate that HBV reactivation associated with ICIs is associated with a high risk of HBV reactivation in HBsAg-positive patients who did not receive prophylactic antiviral therapy.

The mechanism of HBV reactivation by ICIs remains unclear. Immune checkpoint inhibitors have an inhibitory effect on immune tolerance, that is, an immunostimulatory effect. Furthermore, it is thought that HBV reactivation is unlikely to occur. Gane E et al. reported that PD-1 inhibition reduces HBsAg levels [49]. In other words, PD-1 inhibition restored the immune response against HBV-infected hepatocytes, and as a result, the amount of HB surface antigen seems to have decreased. Kamata et al. reported that the PD-1 expression balance between effector T cells and regulatory T cells predicts the clinical efficacy of PD-1 blockade therapy [44]. When PD-1 expression on CD8-positive T cells exceeds that on regulatory T cells, inhibition of immune tolerance by anti-PD-1 antibody administration is expected to have an antitumor effect. On the other hand, it has been reported that when PD-1 expression on regulatory T cells exceeds that on CD8+ T cells, immune tolerance is induced. Furthermore, the antitumor effect is attenuated. Although this report is a study in the cancer-immune environment, it is also an interesting report to consider the mechanism of HBV reactivation by ICIs.

### 1.9. The Latest Treatment Using ICIs: About Atezolizumab/Bevacizumab Curative (ABC) Conversion

Curative treatment (ABC conversion) after the combined use of atezolizumab/bevacizumab for unresectable and transarterial chemoembolization (TACE)-inappropriate intermediate-stage HCC is expected to progress to a radical cure at a high rate [50,51,52,53]. We review the adaptation and timing of curative conversions.

(1)Curative conversion after tumor shrinkage.

Radiofrequency ablation (RFA) and surgery are performed in patients with tumor shrinkage after the atezolizumab/bevacizumab combination.

(2)Curative conversion when tumor shrinkage is not obtained.

Atezolizumab/bevacizumab combination therapy yields a response within four cycles in 80% of responders. Conversely, about 20% of cases show tumor shrinkage after four cycles [54]. Therefore, in patients with advanced-stage HCC, we aim to continue long-term drug administration while well-controlling adverse events (AEs). On the other hand, intermediate-stage HCC can be treated with intensive local therapy such as TACE or RFA. Therefore, if there is no response after four cycles and stable disease or slow progressive disease, rather than continuing drug treatment aimlessly, TACE/RFA is performed to release tumor antigens. Subsequently, atezolizumab/bevacizumab combination therapy is repeated.

(3)TACE-unsuitable nodules other than high tumor burden (multinodular, poorly differentiated HCC, amongst others).

In addition to high tumor burden, TACE-unsuitable patients include poorly differentiated HCC, multinodular HCC, and diffuse HCC [55,56]. Transarterial chemoembolization and RFA are technically possible for such nodules. However, it is not an appropriate treatment option from an oncological point of view. For example, multinodular HCC has a very low capsule formation rate of 5.3% and is TACE-resistant [57]. Microscopically, portal vein invasion and hepatic vein invasion are often observed, and intrahepatic microsatellite is also present [58]. Therefore, recurrence after TACE or RFA is desperate. In such cases, prior drug therapy containing anti-VEGF activity leads to the normalization of tumor blood vessels [34], followed by TACE, RFA, and excision to lead to a radical cure.

(4)Curative conversion during withdrawal due to AE.

It is a treatment method that aims at an abscopal effect by performing selective TACE only on some tumors instead of leaving the tumor until the AE recovers when drug interruption is forced due to AE, such as proteinuria. [57]. That is, intermediate-stage HCC has a relatively long overall survival and, accordingly, a very long treatment period. Therefore, once proteinuria appears in patients with such a long treatment journey, it becomes difficult to use subsequent VEGF inhibitors. In order to avoid this, it is important to reduce the tumor burden and promote the release of tumor antigens by local treatment during drug withdrawal.

(5)Curative conversion for positron-emission tomography (PET)-positive HCC.

Positron emission tomography-positive HCC is basically poorly differentiated liver cancer [59,60]. Additionally, it is biologically aggressive and often recurs after resection, ablation, TACE, or transplantation [61,62]. The effectiveness of ABC conversion has also been reported in such patients [55]

### 1.10. Mechanism of Immune-Related Adverse Effects (irAE) Onset by ICIs

The incidence of liver injury due to ICIs was reported as 6.4% for PD-1 and 7.1% for CTLA-4. However, it increased to 30% when combined [63]. In addition, it is said that the onset of symptoms often occurs 8 to 12 weeks after the start of ICI administration [64]. Hagiwara et al. compared autoimmune hepatitis (AIH) and graft-versus-host disease (GVHD) concerning immunopathology due to irAE liver injury [65]. The CD4/CD8 ratio in irAE cases was 0.52, which was significantly lower (*p* = 0.022) compared to 0.92 in AIH cases. Thus, CD8-positive lymphocyte infiltration was dominant in irAE compared with AIH, especially in the hepatic lobule center. Furthermore, in GVHD cases, the CD4/CD8 ratio was even lower at 0.32, indicating that the effect of CD8-positive cells was even greater in GVHD. As for the onset mechanism of AIH, especially in type II AIH, analysis of antiliver kidney microsome-1 antibody has revealed that the target antigen is cytochrome p450 2D6, which is the cytochrome family of p450 in the liver. On the other hand, although disease-specific antigens have not been identified for Type I AIH, which is common in Japan, there are several candidate antigens, such as antismooth muscle antibodies and soluble liver antigen antibodies. In other words, antigen-presenting cells are sensitized to foreign antigens, such as drugs and pathogens, presented to specific human leucocyte antigen (HLA) class II to activate Helper T cells. Furthermore, CTLs and B are thought to damage cells. In other words, the antigen–antibody reaction is the main cause of AIH development. Consequently, it is thought that the proportion of CD4-positive cells involved is high. On the other hand, in GVHD, it is believed that CD8-positive cells act directly on hepatocytes and bile duct cells that express HLA class I molecules to induce apoptosis. In our study, CD4/CD8 levels were significantly lower in irAE cases than in AIH cases. However, they are not significantly different from GVHD. In irAE, the activated CD8 lymphocytes that lead to the expression of PD-1 are already pooled in the body. Consequently, in this state, the PD-1 pathway is blocked. Based on these immune responses, irAE can be considered GVHD-like rather than AIH-like.

We also examined the degree of Treg infiltration by forkhead box protein 3 (FOXP3) staining. Interestingly, compared with AIH cases, irAE and GVHD cases had significantly lower FOXP3 infiltration, even after correction for CD3. Dysfunction and decreased number of Tregs have been reported to be involved in the pathogenesis of AIH. However, we found that Tregs were further suppressed in irAE compared to AIH. Various reports have been made on the mechanism by which Tregs are suppressed in irAE. In a study using PD-1-deficient mice, Asano et al. showed that isolated Tregs from PD-1-intact mice could migrate from PD-1 intact mice in the presence of low doses of IL-2, a cytokine important for Treg expansion [66]. Thus, they reported that the systemic administration of PD-1 antibody reduced the number of Tregs by enhancing apoptosis. In addition, Kido et al. created a hepatitis model using athymic/PD-1 knockout mice and reported that in this model, CD8-positive T cells infiltrate the lobular at the early stage of inflammation [67]. This observation was very similar to the liver pathology in our irAE patient here, suggesting that the decrease in Tregs is involved in the onset and pathogenesis of irAE. Furthermore, cytokines and chemokines produced by tumor cells are involved in the differentiation, survival, and invasion of suppressive immune cells such as Tregs and TMA. Therefore, not only PD-1 antibody administration but also PD-L1 antibody administration seems to cause a decrease in Tregs. In fact, in this study, Tregs were suppressed not only in PD-1-administered patients but also in PD-L1 patients. Furthermore, in GVHD, as in irAE, the infiltration of FOXP3-positive cells was lower than in AIH. Although it is not clear how the reduction in Treg in GVHD affects pathology, it was reduced similarly to irAE. Immune dysregulation, polyendocrinopathy, enteropathy, X-linked syndrome is a representative disease in which Tregs are suppressed. It is a group of diseases caused by mutations in FOXP3 that cause regulatory T cell dysfunction and subsequent autoimmune diseases. In addition to hepatitis, endocrine disorders such as thyroid and diabetes, and enteritis associated with protein leaks, abnormalities in the blood cell system, skin diseases, kidney diseases, and various other organs are damaged. Compared to AIH, irAE and GVHD can be assumed to be due to the simultaneous or metachronous severe damage to multiple organs.

### 1.11. Classifications of Tumor Microenvironment (TME) Factors

The integrative analysis of tumor microenvironment components and their microanatomical distribution help to comprehend complexity of cell progression and differentiation. Heterogeneity of HCC has an impact on efficacy of clinical treatments. Even it is not clearly investigated, understanding heterogeneity is critical to improve drug response, such as atezolizumab and bevacizumab combination therapy and clinical outcome of immune checkpoint inhibitors. Therefore, it is important to take into consideration the spatial biology of HCC and recently developed spatial DNA/RNA sequencing to configurate an impact of immunosuppressive activities across the tumor, nontumor, and peripheral blood microenvironments. It has been shown that cytometry by time-of-flight for characterization (CyTOF) is useful for the characterization of the cancer-immune landscape in HCC [68]. CyTOF systems enable configurating an impact of immunosuppressive activities across the tumor, nontumor, and peripheral blood microenvironments. Moreover, CyTOF can be applied to anticipate patients who are treated immunosuppressive agents into responders and nonresponders [69]. Although CyTOF has limitations, such as the number of the markers and the need to focus cell types [70,71], studies using CyTOF have provided an in-depth understanding of the tissue microenvironment of HCC progression as biomedical research translates spatial genomic data into differentiation of multiple therapeutical response [72]. Therapeutic response rates with specific molecular targets or immune modulation are quite different, even in the same pathological type and clinical stage. These clinical challenges suggest that HCC patients need to be precisely at the molecular level. To uncover the biological events happening in HCC at a high resolution, single-cell RNA sequencing would make it possible to gain quantitative transcriptomic information at a single-cell level, at the same time eradicating the bias occurring in bulk-RNA sequences caused by different cell populations [73,74,75]. Adapting single-cell RNA sequencing relies on complex interaction, representing collective efforts of physicians from genetics, biomedical engineering, statistics, and bioinformatics [76].

## 2. Conclusions

New findings are elucidating the complex immune microenvironment of HCC. It is also important that the tumor immune environment can also be altered by etiology, such as NASH. In the future, it is expected to be applied to the optimization of treatment according to such tumor immune microenvironment.

In addition, when administering immunotherapy, the timing of treatment and combination with local therapies such as RFA and TACE are particularly important. It is necessary to be aware that this may significantly change the therapeutic effect and prognosis. Furthermore, the management of its side effects, such as HBV reactivation, is also important.

Further advances in immunotherapy for HCC are expected in the future.

## Figures and Tables

**Figure 1 cancers-15-02070-f001:**
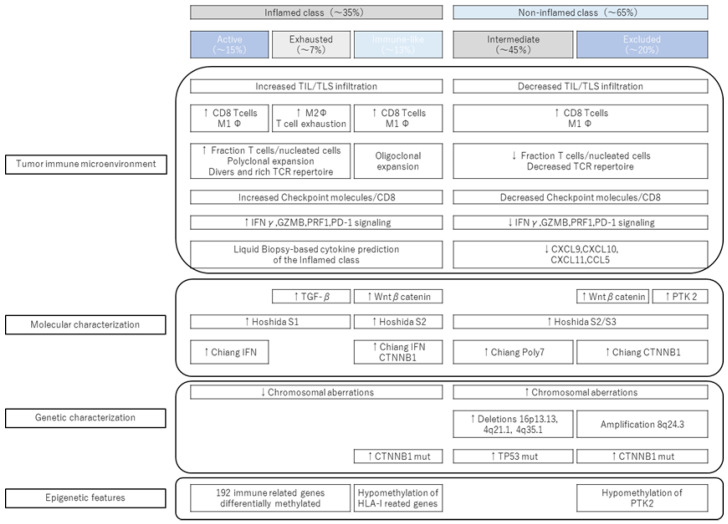
New immune subclass (modified quotation from [17]). A new immune-like subclass was proposed as a subclass of the inflamed class. While having similar immunological characteristics to the immune active/immune exhausted subclass, there are differences in molecular characteristics.

**Figure 2 cancers-15-02070-f002:**
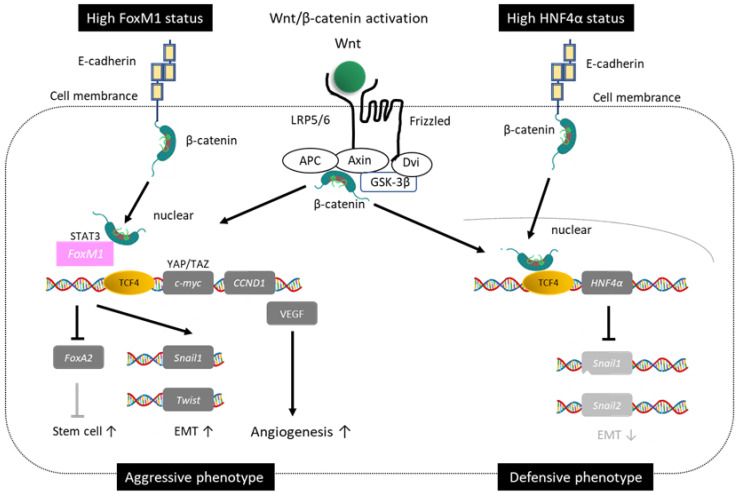
Duality of Wnt/β-catenin mutation (modified from [18]). Wnt/β-catenin mutations in HCC are divided into favorable and unfavorable phenotypes. HCC-specifically expressed tumor suppressor gene HNF4α is a target gene of Wnt/β-catenin signaling.

**Figure 3 cancers-15-02070-f003:**
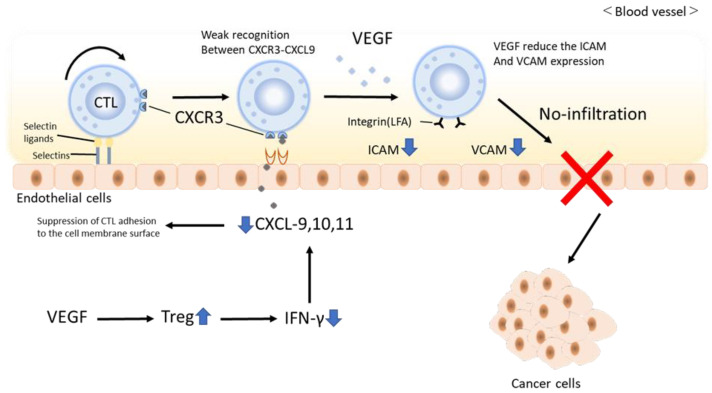
Mechanism of suppression of tumor infiltration of CD8-positive T cells by VEGF. In the cancer immune cycle, VEGF prevents infiltration of CD8-positive T cells flowing in the blood into the tumor. The main factors are that VEGF induces FAS ligand and causes CTL to undergo apoptosis, and that by reducing adhesion factors such as ICAM-1 and VCAM-1, CTL adheres to the vascular endothelium and migrates to tumors. known to reduce performance. Furthermore, blocking the release of chemokines such as CXCL9, 10, and 11, which are ligands of CXCR3, from intratumoral dendritic cells prevents CTL from adhering to vascular endothelium and inhibits intratumoral invasion.

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
