# Peer review of "Advances in Immunotherapy for Hepatocellular Carcinoma"

_cancers, 2023, doi:10.3390/cancers15072070_

Round 1
Reviewer 1 Report
1) The abstract is very concise. Authors should add hints on the topics that will be discussed later in the review.
2) L52: Delete abbreviation in subtitles "Immune subclass of hepatocellular carcinoma (HCC)"
3) L58: Write all abbreviated forms of genes in italics "PD-1" to differentiate them from the proteins.
4) Summarize all Immune subclass of HCC in a table.
5) L66: Check this abbreviation "Catenin Beta 1 (CTNNB1)"
6) L97-104: Add References.
7) L107-109: check for proper grammar and language.
8) L106: It would be better to add a diagram explaining "Duality of Wnt/β-catenin mutations"
9) L148: "NASH" Add full term at first mention then use the abbreviated form. A short introduction (Definition) of NASH-HCC is also wanted in the first sentence.
10) L150: "cancer surface cancer" Check for language.
11) L165: "immunotherapy (ICI)" incorrect abbreviation.
12) L173-193: It would be better to add a diagram to explain the mode of action of atezolizumab and bevacizumab.
13) It is not clear whether the data presented in Figures 1 and 2 belong to Hagiwara et al. or yours. So either add copyright privacy permission or indicate the Method used to get these data.
14) In general the titles are not well organized and some of them should be combined, see for example:
Combine "Duality of Wnt/β-catenin mutations" with"Gadoxetic acid-enhanced magnetic resonance imaging (EOB-MRI) as an imaging biomarker for Wnt/β-catenin mutations"
Also combine "Effects of immunotherapy on NASH-HCC" with "Background of NASH-HCC: the relationship between progression of liver fibrosis and genomic and epigenomic mutations"
15) L256: "Furthermore, we have also reported that PD-1 expression in tumor local Tregs is a useful biomarker for PD-1/PD-L1 inhibitor therapy" Are these published data or what? Add references.
16) L350: Replace "We" with "author names/citation"
17) Conclusion is so poor and less informative.
18) This review lacks information on other immunotherapy drugs approved by FDA such as durvalumab, tremelimumab, pembrolizumab, nivolumab, and ipilimumab. either as individual therapy or in combination.
Author Response
Point-by-Point Response to the Editor’s and Reviewer’s comments:
We appreciate the Reviewer’s positive comments regarding our manuscript. We have studied the comments carefully and addressed the queries as shown below.
Reviewer 1
Queries 1:
The abstract is very concise. Authors should add hints on the topics that will be discussed later in the review.
Response:
Thank you for your valuable suggestion. Added topics to the abstract.
Queries 2:
L52: Delete abbreviation in subtitles "Immune subclass of hepatocellular carcinoma (HCC)"
Response:
Thank you for your valuable suggestion. Deleted as you suggested.
Queries 3:
L58: Write all abbreviated forms of genes in italics "PD-1" to differentiate them from the proteins.
Response:
Thank you for your valuable suggestion. Fixed as you pointed out.
Queries 4:
Summarize all Immune subclass of HCC in a table.
Response:
Thank you for your valuable suggestion. As you pointed out, I added a figure.
Queries 5:
L66: Check this abbreviation "Catenin Beta 1 (CTNNB1)"
Response:
Thank you for your valuable suggestion. Fixed as you pointed out.
Queries 6:
L97-104: Add References.
Response:
Thank you for your valuable suggestion. Added as you pointed out.
Queries 7:
L107-109: check for proper grammar and language.
Response:
Thank you for your valuable suggestion. Fixed as you pointed out.
Queries 8:
L106: It would be better to add a diagram explaining "Duality of Wnt/β-catenin mutations"
Response:
Thank you for your valuable suggestion. I added a figure as you suggested.
Queries 9:
L148: "NASH" Add full term at first mention then use the abbreviated form. A short introduction (Definition) of NASH-HCC is also wanted in the first sentence.
Response:
Thank you for your valuable suggestion. Fixed.
Queries 10:
L150: "cancer surface cancer" Check for language.
Response:
Thank you for your valuable suggestion. Fixed.
Queries 11:
L165: "immunotherapy (ICI)" incorrect abbreviation.
Response:
Thank you for your valuable suggestion. Fixed.
Queries 12:
L173-193: It would be better to add a diagram to explain the mode of action of atezolizumab and bevacizumab.
Response:
Thank you for your valuable suggestion. I added a figure as you suggested.
Queries 13:
It is not clear whether the data presented in Figures 1 and 2 belong to Hagiwara et al. or yours. So either add copyright privacy permission or indicate the Method used to get these data.
Response:
Thank you for your valuable suggestion. The discussion of methylation is not sufficient and has been removed from this paper.
Queries 14:
In general the titles are not well organized and some of them should be combined, see for example:
Combine "Duality of Wnt/β-catenin mutations" with"Gadoxetic acid-enhanced magnetic resonance imaging (EOB-MRI) as an imaging biomarker for Wnt/β-catenin mutations"
Also combine "Effects of immunotherapy on NASH-HCC" with "Background of NASH-HCC: the relationship between progression of liver fibrosis and genomic and epigenomic mutations"
Response:
Thank you for your valuable suggestion. As you pointed out, we merged. Also, since the discussion of methylation is not sufficient, it has been removed from this paper.
Queries 15:
L256: "Furthermore, we have also reported that PD-1 expression in tumor local Tregs is a useful biomarker for PD-1/PD-L1 inhibitor therapy" Are these published data or what? Add references.
Response:
Thank you for your valuable suggestion. I added the ref as you pointed out.
Queries 16:
L350: Replace "We" with "author names/citation"
Response:
Thank you for your valuable suggestion. Fixed as you pointed out.
Queries 17:
Conclusion is so poor and less informative.
Response:
Thank you for your valuable suggestion. Added conclusion.
Queries 18:
This review lacks information on other immunotherapy drugs approved by FDA such as durvalumab, tremelimumab, pembrolizumab, nivolumab, and ipilimumab. either as individual therapy or in combination.
Response:
Thank you for your valuable suggestion.  In Japan, atezolizumab is used for HCC. Only MSI-high pembrolizumab has been used, but very rarely. Therefore, we will focus on atezolizumab this time.
Reviewer 2 Report
Revolutionary advances in immune checkpoint blockade (ICB) therapy, such as anti-PD-1/L1 and anti-CTLA4 antibodies, have extended patient survival in multiple cancers. Although many efforts have been made to improve the clinical benefit of HCC immunotherapy, especially T cells with different cytolytic activities, the response to ICB in HCC patients remains limited.
In this manuscript, the authors reviewed the current status of immunotherapy in HCC and updated the classifications of tumor microenvironment (TME) factors. Although authors pointed out the critical role of TME in the response of ICB therapy, they did not discuss at the molecular level which factors of the TME could hinder the ICB response. Authors should provide a detailed update of recent development in the field of spatial biology of HCC that uncovers the complex heterogeneity of tumor cells and TME and factors that may hinder the ICB response.
Author Response
Reviewer 2’s comment:
Revolutionary advances in immune checkpoint blockade (ICB) therapy, such as anti-PD-1/L1 and anti-CTLA4 antibodies, have extended patient survival in multiple cancers. Although many efforts have been made to improve the clinical benefit of HCC immunotherapy, especially T cells with different cytolytic activities, the response to ICB in HCC patients remains limited.
In this manuscript, the authors reviewed the current status of immunotherapy in HCC and updated the classifications of tumor microenvironment (TME) factors. Although authors pointed out the critical role of TME in the response of ICB therapy, they did not discuss at the molecular level which factors of the TME could hinder the ICB response. Authors should provide a detailed update of recent development in the field of spatial biology of HCC that uncovers the complex heterogeneity of tumor cells and TME and factors that may hinder the ICB response.
Response:
Thank you for your valuable suggestion. I have added the content you pointed out to the text. Please forgive me for any inadequacies.
Reviewer 3 Report
This is a good review paper to summary the immunotherapy for HCC. The authors discussed the disease subclasses/ classifications, Wnt/beta-catenin pathway, high frequent mutation, methylation and virus reactivation. However, I have several major concerns for this review paper. (1) The structure of the paper is not very clear. (2) Insufficient legend for the figures. (3) The authors only report the result from one or few studies to support the conclusion. But study with larger sample size or multiple studies are expected to draw robust conclusions in a review paper. (4) English writing to be improved. Specifically,
1. It’s unclear how are HCC cases were divided by inflamed class and non-inflamed class. By which features or diagnosis?
2. Exon 3 mutation of CTNNB1 shows more aggressive phenotype. Can the author specifically discuss this when introducing the CTNNB1 mutation?
3. Can the authors add in the crosstalk between CTNNB1 mutation and other oncogenes that can induce tumor?
4. What does NSG short for (in line 201 and 415)?
5. Figure 1:
a. Each row is a gene. But in the legend, it says ‘Each row corresponds to an individual case’. It should be ‘col’ for individual case, right?
b. We can see blue and red colors in the heatmap, but the color key only shows gradient orange colors?
6. In order to illustrate the high-frequent mutated genes in HCC patients, I would suggest to explore the cBioportal (https://www.cbioportal.org/) or TCGA database, which have much larger sample size. Only 8 vs 8 cases in Figure 1 is not sufficient to claim the mutation frequency.
7. Which methylation study did the author explore? Not citation at all. And is there any study with larger sample size that can be explore to draw the conclusion? This is a review paper, so I would expect the authors can cite larger study, or combine multiple studies to draw the conclusion.
8. Figure 2: no legend for red and green colors. In Figure 2 legend, it says the principal component analysis shows p=0.0007, but in the figure it says by Fisher exact test. The p-value is so confusing. Which test is it used? Is it for a specific gene or the authors pool all the genes together?
9. Cannot understand what the authors want to expression for line 272-275. “In contrast, one of fifty-five (1.8%) HBV DNA-negative patients received prophylactic antiviral therapy. Furthermore, HBV DNA-negative patients received prophylactic antiviral therapy. Reactivation was seen in 5 of 24 patients (20.8%) who did not receive antiviral therapy for 6 patients (7.5%).”
Author Response
Reviewer 3’s comment:
This is a good review paper to summary the immunotherapy for HCC. The authors discussed the disease subclasses/ classifications, Wnt/beta-catenin pathway, high frequent mutation, methylation and virus reactivation. However, I have several major concerns for this review paper. (1) The structure of the paper is not very clear. (2) Insufficient legend for the figures. (3) The authors only report the result from one or few studies to support the conclusion. But study with larger sample size or multiple studies are expected to draw robust conclusions in a review paper. (4) English writing to be improved.
Queries 1:
It’s unclear how are HCC cases were divided by inflamed class and non-inflamed class. By which features or diagnosis?
Response:
Thank you for your valuable suggestion.  Clarified how to classify subclasses.
Queries 2:
Exon 3 mutation of CTNNB1 shows more aggressive phenotype. Can the author specifically discuss this when introducing the CTNNB1 mutation?
Response:
Thank you for your valuable suggestion.  Added.
Queries 3:
Can the authors add in the crosstalk between CTNNB1 mutation and other oncogenes that can induce tumor?
Response:
Thank you for your valuable suggestion.  Added.
Queries 4:
What does NSG short for (in line 201 and 415)?
Response:
Thank you for your valuable suggestion.  Added.
Queries 5:
Figure 1:
- Each row is a gene. But in the legend, it says ‘Each row corresponds to an individual case’. It should be ‘col’ for individual case, right?
- We can see blue and red colors in the heatmap, but the color key only shows gradient orange colors?
Response:
Thank you for your valuable suggestion. The discussion of methylation is not sufficient and has been removed from this paper.
Queries 6:
In order to illustrate the high-frequent mutated genes in HCC patients, I would suggest to explore the cBioportal (https://www.cbioportal.org/) or TCGA database, which have much larger sample size. Only 8 vs 8 cases in Figure 1 is not sufficient to claim the mutation frequency.
Response:
Thank you for your valuable suggestion. The discussion of methylation is not sufficient and has been removed from this paper.
Queries 7:
Which methylation study did the author explore? Not citation at all. And is there any study with larger sample size that can be explore to draw the conclusion? This is a review paper, so I would expect the authors can cite larger study, or combine multiple studies to draw the conclusion.
Response:
Thank you for your valuable suggestion. The discussion of methylation is not sufficient and has been removed from this paper.
Queries 8:
Figure 2: no legend for red and green colors. In Figure 2 legend, it says the principal component analysis shows p=0.0007, but in the figure it says by Fisher exact test. The p-value is so confusing. Which test is it used? Is it for a specific gene or the authors pool all the genes together?
Response:
Thank you for your valuable suggestion. The discussion of methylation is not sufficient and has been removed from this paper.
Queries 9:
Cannot understand what the authors want to expression for line 272-275. “In contrast, one of fifty-five (1.8%) HBV DNA-negative patients received prophylactic antiviral therapy. Furthermore, HBV DNA-negative patients received prophylactic antiviral therapy. Reactivation was seen in 5 of 24 patients (20.8%) who did not receive antiviral therapy for 6 patients (7.5%).”
Response:
Thank you for your valuable suggestion.  Corrected the text.
In addition, we have added some additional references to support the revised section in the manuscript.
Round 2
Reviewer 1 Report
Authors responded to all my comments.
Author Response
Reviewer 1’s comment:
Authors responded to all my comments.
Response:
Thank you for your valuable suggestion. Thank you very much.
Reviewer 2 Report
In the revised manuscript authors added more details about the classifications of tumor microenvironment (TME) factors. However they did not fully addressed my concerns in the first version. Here, authors should write a new paragraph about the spatial biology of HCC and recently developed spatial DNA/RNA sequencing to identify subtumor populations and stroma components that are associated with ICB response.
Author Response
Reviewer 2’s comment:
In the revised manuscript authors added more details about the classifications of tumor microenvironment (TME) factors. However they did not fully addressed my concerns in the first version. Here, authors should write a new paragraph about the spatial biology of HCC and recently developed spatial DNA/RNA sequencing to identify subtumor populations and stroma components that are associated with ICB response.
Response:
Thank you for your valuable suggestion. Created a new paragraph.
Reviewer 3 Report
The authors have addressed most of my concerns. Thanks for the great work.
My major concern is that, the three new figures are kind of the same as the previous published work. The review paper is expected to integrate multiple works and generate new summary figure. If there is nothing new to be added to the previous published work, the author should just cite it instead of replot it.
1/ The new Figure 1: Though the authors cited reference 16 and 17, the newly added figure is almost the same content as these two reference manuscripts, except for taking out some images for the staining slides. I would suggest the authors re-generate a new summary figure by reviewing multiple manuscripts and integrate their ideas together, rather than exact copy of the published figure.
2/ The new figure 2: Though the authors cited reference 18, this new figure is exactly the same as the Figure 3 of reference 18. The authors just replotted this figure but with exactly the same content and color. I would expect a review paper to summarize the previous research, but not copy the same figure. If there is no new content, the author can just cite reference 18. Why replot it?
3/ the new figure 3: This figure is very similar to the figure where the corresponding author has published in 2022: https://www.karger.com/Article/Abstract/524977. If the content is the same, please cite the previous publication, rather than replot it.
Author Response
Reviewer 3’s comment:
The authors have addressed most of my concerns. Thanks for the great work.
My major concern is that, the three new figures are kind of the same as the previous published work. The review paper is expected to integrate multiple works and generate new summary figure. If there is nothing new to be added to the previous published work, the author should just cite it instead of replot it.
Queries 1:
1/ The new Figure 1: Though the authors cited reference 16 and 17, the newly added figure is almost the same content as these two reference manuscripts, except for taking out some images for the staining slides. I would suggest the authors re-generate a new summary figure by reviewing multiple manuscripts and integrate their ideas together, rather than exact copy of the published figure.
Response:
Thank you for your valuable suggestion. I searched again, but reference 16 and 17 were the latest information. Therefore, I have clarified that I have modified from reference 16.17 in the paper.
Queries 2:
2/ The new figure 2: Though the authors cited reference 18, this new figure is exactly the same as the Figure 3 of reference 18. The authors just replotted this figure but with exactly the same content and color. I would expect a review paper to summarize the previous research, but not copy the same figure. If there is no new content, the author can just cite reference 18. Why replot it?
Response:
Thank you for your valuable suggestion. Modified Figure2. In addition, I have clarified that I have modified from reference 18 in the paper.
Queries 3:
3/ the new figure 3: This figure is very similar to the figure where the corresponding author has published in 2022: https://www.karger.com/Article/Abstract/524977. If the content is the same, please cite the previous publication, rather than replot it.
Response:
Thank you for your valuable suggestion. Modified Figure3. In addition, I have clarified that I have modified from reference 41 in the paper.
In addition, we have added some additional references to support the revised section in the manuscript.
Round 3
Reviewer 3 Report
I’m still concerning about the figures that are highly similar to the previous publications.